# Changes in the Lifestyle of the Spanish University Population during Confinement for COVID-19

**DOI:** 10.3390/ijerph19042210

**Published:** 2022-02-15

**Authors:** Sandra Sumalla-Cano, Tamara Forbes-Hernández, Silvia Aparicio-Obregón, Jorge Crespo, María Eléxpuru-Zabaleta, Mónica Gracia-Villar, Francesca Giampieri, Iñaki Elío

**Affiliations:** 1Research Group on Foods, Nutritional Biochemistry and Health, Universidad Europea del Atlántico, 39011 Santander, Spain; sandra.sumalla@uneatlantico.es (S.S.-C.); maria.elexpuru@uneatlantico.es (M.E.-Z.); 2Department of Health, Nutrition and Sport, Universidad Internacional Iberoamericana, Campeche 24560, Mexico; silvia.aparicio@uneatlantico.es (S.A.-O.); jorge.crespo@uneatlantico.es (J.C.); monica.gracia@uneatlantico.es (M.G.-V.); 3Biomedical Research Centre, Department of Physiology, Institute of Nutrition and Food Technology ‘‘José Mataix”, University of Granada, 1871 Granada, Spain; tamaraforbes@gmail.com; 4Faculty of Social Sciences and Humanites, Universidad Europea del Atlántico, 39011 Santander, Spain; 5Higher Polytechnic School, Universidad Europea del Atlántico, 39011 Santander, Spain; 6Department of Biochemistry, Faculty of Sciences, King Abdulaziz University, Jeddah 21589, Saudi Arabia

**Keywords:** COVID-19, eating habits, physical activity, MEDAS-14, Emotional Eating Questionnaire (EEQ) and university population

## Abstract

The aim of this study was to evaluate the influence of the lockdown due to the COVID-19 pandemic, on eating and physical activity behavior, in a university population. A healthy diet such as the Mediterranean Diet (MD) pattern, rich in fruit and vegetables, can prevent degenerative diseases such as obesity, diabetes, cardiovascular diseases, etc. We conducted a cross-sectional study and data were collected by an anonymous online questionnaire. Participants completed a survey consisting of 3 sections: sociodemographic data; dietary behavior and physical activity; the Mediterranean Diet questionnaire (MEDAS-14) and the Emotional Eater Questionnaire (EEQ). A total of 168 participants completed the questionnaire: 66.7% were women, 79.2% were from Spain, 76.8% were students, 76.2% lived in their family home and 66.1% were of normal weight. During lockdown our population shopped for groceries 1 time or less per week (76.8%); maintained the same consumption of fruits (45.2%), vegetables (50.6%), dairy products (61.9%), pulses (64.9%), fish/seafood (57.7%), white meat (77.4%), red and processed meat (71.4%), pastries and snacks (48.2%), rice/pasta/potatoes (70.2%) and nuts (62.5%), spirits (98.8%) and sugary drinks (91.7%). Cooking time increased (73.2%) and the consumption decreased of low alcohol drinks (60.1%), spirits (75%) and sugary drinks (57.1%), and physical activity also diminished (49.4%). University Employees (UE) gained more weight (1.01 ± 0.02) than students (0.99 ± 0.03) (*p* < 0.05) during the confinement period. A total of 79.8% of the participants obtained a Medium/High Adherence to the MD during lockdown. Emotional and very emotional eaters were higher in the female group (*p* < 0.01). In the event of further confinement, strategies should be implemented to promote a balanced and healthy diet together with the practice of physical activity, taking special care of the female and UE groups.

## 1. Introduction

At the end of 2019, a new highly infectious and pathogenic strain of RNA viruses belonging to the Coronaviridae family emerged in Wuhan, China [1]. ‘Coronavirus disease 2019′ (COVID-19) showed high mortality rates and a rapid human-to-human worldwide spread that switched into a global health emergency a few months after its outbreak, leading the World Health Organization (WHO) to declare it a pandemic in March 2020 [2]. Every country in the world confirming COVID-19-positive cases established strict preventive health measures to avoid viral transmission, including exhaustive testing, case tracing, and severe social restrictions. However, these limitations showed poor efficacy against infection; thus, governments were finally forced to enforce mandatory quarantine at home [3]. Due to the gravity of the situation in Spain, a State of Alarm was declared on the 24 March 2020 [4], which, for the population, resulted in serious social restrictions. During this period, the Spanish people were confined to their homes and were allowed to leave them only to perform very limited essential activities. This confinement lasted until the 2 May 2020. From that moment on, the population was allowed to leave their homes, their presence being regulated on public roads by age and time of day [5].

In the case of university education, at first it was suspended for a few days or weeks, depending on the decision of each institution. Subsequently, and with the extension of confinement, the majority chose to continue teaching online until the following academic year. In the case of the Universidad Europea del Atlántico, teaching online was carried out sequentially subject by subject with in-person optional reinforcement classes in June and ordinary face-to-face exams in July and September.

These rigorous social measures affected the lifestyle habits of the population, mostly those related to eating patterns and physical activity [6]. It is worth mentioning that this imposed isolation at home was considered a stressful situation that further influenced diet routines since people cooked more and consumed more appetizing meals, snacks or alcohol [7,8,9,10,11]. With the intention to avoid a poor nutritional diet during lockdown, the WHO published dietary guidelines highlighting the need for balanced diet patterns. In these guidelines the WHO advise the consumption of 9 fruit and vegetable servings/day (4 and 5 servings/day respectively) together with legumes, meat, and foods made from whole grain cereals [2]. It is important to note that these dietary recommendations describe all the main nutritional elements of the Mediterranean diet (MD) [11,12].

It is currently a fact that the MD is one of the most beneficial dietary patterns known in the world [13,14,15]. Fruits, vegetables, nuts, legumes, whole grains and olive oil are the most representative foods of this diet rich in bioactive components such as polyphenols (the main group of plant-derived metabolites [16]), which confer high anti-inflammatory, antioxidant, antimicrobial, immunomodulatory, antiviral or neuroprotective beneficial properties [17,18,19,20,21]. Thanks to these healthy phytochemicals, MD has been widely associated with a lower risk of developing chronic inflammatory diseases and related comorbidities (obesity, type 2 diabetes and metabolic syndrome), cancer or age-associated disorders among others [14,15].

In the current pandemic situation, the lockdown imposed by governments due to the spread of SARS-Cov-2 has critically influenced our lifestyle habits. Numerous studies from many countries and regions around the world have analyzed the consequences of quarantine on our health, with diverse outcomes depending on the adherence to a specific diet pattern and physical activity schedule [7,11,22,23,24,25,26,27,28,29]. However, very few have focused on studying the consequences of this situation in the Spanish university population [30,31], who, in addition to the stress caused by the global pandemic, had to continue with their studies, writing papers and preparing for final exams. In addition, the novelty of our study consists in analyzing the consequences of this situation in the university staff who had to readapt their work and teaching to the virtual environment in real time. To the best of our knowledge, very few studies have investigated the effect of confinement on emotional eating behavior [32]. In this context, we aimed to evaluate whether the quarantine associated with the COVID-19 pandemic situation has influenced or modified the dietary habits (specifically the adherence to the MD) and emotional eating behavior and physical activity practices of a population comprising staff and students belonging to the Universidad Europea del Atlántico.

## 2. Materials and Methods

### 2.1. Selection of Participants and Study Design

A cross-sectional descriptive study based on a self-administered questionnaire was carried out. It was a non-probabilistic sample used for convenience since the questionnaire was directed towards people of legal age (≥18 years old) from the university community recruited online. The collection of information began on 28 April 2020; 46 days after lockdown had begun (Phase 1) and the last data were sent on 29 May 2020.

For a correct collection of the information, a Google form application which allows the anonymity of the participants was used. The questionnaire was directed towards the entire university community (students, teachers and administrative staff), by email. After agreeing to complete the questionnaire, participants were asked to answer the questions of the self-administered online questionnaire, with all answers mandatory, (in a time of less than 30 min) informing them that they could interrupt the compilation at any time, without the obligation of justifying their decision. Participation in the study was voluntary and completely free. 

The study was approved by the Committee of Ethics of Investigation of the Universidad Europea del Atlántico (C-12/2020), and all the anonymously compiled information was treated with maximum confidentiality in accordance with the law for Protection of Personal Data and guarantee of digital rights 3/2018 of 5 December 2018 [33]. The data collected will only be used for research purposes following the scientific method, respecting the Regulation (EU) of the European Parliament and of the Advice 2016/679 of 27 April 2016, relative to the protection of natural persons regarding the processing of personal data and to the free movement of such data, and repealing Directive 95/46/EC (General Data Protection Regulation) [34].

### 2.2. Instrument and Variables

Within the questionnaire, information was organized in 3 sections. A first section (25 items) included the sociodemographic profile: sex, age range, birthplace, cohabitation with other people, place of residence during confinement, position in the University and, for students, the faculty and degree program. They were also asked for their height in meters, their weight before confinement and how many kilograms their weight had changed during confinement. Both height and weight were self-referenced, the participants were asked to include their usual weight before confinement and if this had changed, to indicate the kilograms gained or lost.

Dietary behavior and physical activity habits were described in the second section (17 items) with questions about grocery shopping, number of meals, change in consumption of fruit, vegetables, dairy products, pulses, fish/seafood, white meat, red meat and processed meat, pastries and snacks, rice/pasta/potatoes, nuts, alcoholic drinks and sugary drinks, based on the healthy eating pyramid [35]. To establish changes in consumption, participants were asked whether their consumption of the different food groups had decreased, remained the same or increased. To establish changes in physical activity, participants were also asked whether their activity had decreased, remained the same or increased during confinement. In all cases they were asked to compare their usual habits before confinement with their habits at the current time of confinement.

A third section included a semiquantitative questionnaire of adhesion to the Mediterranean Diet (MD) (MEDAS-14) [36] and a validated questionnaire of emotional eating “The Emotional Eater Questionnaire” (EEQ) [37]. MEDAS-14 was assessed on the continuous scale (0–14 points) classifying the participants into low (<5 points), medium (6 to 8 points) and high (>9 points) adherence levels [36], while EEQ was assessed on the continuous scale (0–30 points), classifying the participants into non-emotional eater (0–5 points), low emotional eater (6–10 points), emotional eater (11–20 points) and very emotional eater (21–30 points) [37]. 

A pilot study was carried out among 15 subjects to verify the effectiveness of the questionnaire to learn if it provided the necessary information or if any of the questions needed to be modified.

A total of 170 completed questionnaires were received and two of them were eliminated because they both consisted of the same data in all the answers. Thus, 168 questionnaires were considered valid.

### 2.3. Statistical Analysis

Normal distribution for quantitative variables was assessed using the Shapiro–Wilk normality test. The quantitative variables were expressed as mean and standard deviation (SD) or as median accompanied by their interquartile range (IQR), as appropriate. Body mass index (BMI) was distributed using WHO ranges classification. 

The following variables were regrouped to establish more robust results: MEDAS-14 (Low adherence and Medium/High), BMI (Under/Normal weight and Pre-obesity/Obesity I and II) and EQQ (Non-emotional/Low emotional eater and Emotional/Very emotional eater). 

For the quantitative variables, the Student’s test or non-parametric Mann–Whitney U test was used. Univariate comparisons were investigated between groups and explicative variables using Pearson X^2^. The criterion of significance was established at *p* < 0.05. All computer data were analyzed using “The jamovi project (2021). *jamovi*. (Version 1.8) [Computer Software]. Retrieved from https://www.jamovi.org (accessed on 12 June 2021).” Epidat 4.2: software for epidemiological data analysis. Version 4.2, July 2016. Consellería de Sanidade, Xunta de Galicia, Spain; Pan American Health Organization (PAHO-WHO); Universidad CES, Colombia, was used to analyze the frequency distribution and confidence intervals for the categorical variables. 

## 3. Results

### 3.1. Characterization of the Sample Population

Two thirds (66.7%) of the participants were women, 79.2% from Spain, students (76.8%), living in the family home (76.2%) and in the range of age of 21–35 years (47.6%). The BMI, calculated with the weight prior to the lockdown, showed that 7.1% were underweight, 66.1% of the sample had a normal weight, while the 26.8% were pre-obese or had some type of obesity. 

Analyzed by role, there were students (79.8%), Research Professors (R.P) (14.9%) and Administrative Staff (A.S) (8.3%). Students were from the Faculty of Health and Science (65.9%), Faculty of Social and Humanity Sciences (27.1%) and Higher Polytechnic School (7%). The degree programs with the greater number of students were Psychology (26.4%) followed by Human Nutrition and Dietetics (19.4%), Sport and Exercise Sciences (9.3%), Food and Sciences Technology (7.8%), Translation and Interpretation (7%), Advertising and Public relations (6.2%), Business Administration and Management (5.4%) and others (18.7%). Completed data can be found in Appendix A.

Analyzing by role of students vs. University Employees (UE) (Table 1), significant differences were found: in the proportion of women, higher in students (71.3%; CI 62.6–78.9) than in UE (51.3%; CI 34.7–67.5) (*p* < 0.05); in age range > 36 years, higher in UE (82.1%; CI 66.4–92.4) than in students (1.6%; CI 0.1–5.4) (*p* < 0.001); and in Pre-obesity (25–29.9 kg/m^2^), higher in UE (35.9%; CI 21.2–52.8) than in students (17.1%; CI 11.0–24.6) (*p* < 0.05).

To analyze weight changes during the lockdown, it was decided to use weight variation (weight increased in kg/weight before de confinement); 1 meant that the current weight was maintained, <1 implied a weight reduction and >1 a weight gain. In general, our population maintained or increased weight (mean of 0.99 ± 0.03). Non-statistically significant differences were found in weight variation comparing women (median: 1.00; IQR: 0.043) and men (median: 1.00; IQR: 0.050) (*p* = 0.507) and comparing the origin of our population between Spain (median: 1.00; IQR: 0.040) and others (median: 1.00; IQR: 0.054) (*p* = 0.493). However, between Students (median: 1.00; IQR: 0.038) and UE (Research Professor and Administrative Staff) (median: 1.01; IQR: 0.029), significant differences were found between the two groups (*p* < 0.05) using the Mann–Whitney U test. (Figure 1).

Specifically for students, non-statistically significant differences were found, comparing the Degree of Human Nutrition (median: 1.00; IQR: 0.016) versus others (median 1.00; IQR: 0.044) (*p* = 0.828) and comparing by Faculty of Health Sciences (median: 1.00; IQR: 0.040) versus others (median: 1.00; IQR: 0.038) (*p* = 0.910) using the Mann–Whitney U test. 

### 3.2. Confinement Effect on Dietary and Physical Activity Behavior

To highlight how lockdown affected the dietary behavior of the population, individuals were questioned about the changes in their habits and dietary pattern compared to the preceding period (Table 2). Completed data can be found in Appendix A.

During lockdown, most of the evaluated population did the grocery shopping once or less per week (76.8%) and cooked more than usual (73.2%). Consumption remained the same of fruit (45.2%; CI 37.5–53.0), vegetables (50.6%; CI 42.7–58.3), dairy products (61.9%; CI 54.1–69.2), pulses (64.9%; CI 57.1–72.0), fish/seafood (57.7%; CI 49.8–65.3), white meat (77.4%; CI 70.3–83.4), red and processed meats (71.4%; CI 63.9–78.1), pastries and snacks (48.2%; CI 40.4–56.0), rice/pasta/potatoes (70.2%; CI 62.7–77.0), nuts (62.5%; CI 54.7–69.8) and the number of meals (47.6%; CI 39.8–55.4). Decreased consumption was found of low alcohol drinks (60.1%; CI 52.2–67.5), spirits (75%; CI 67.7–81.3) and sugary drinks (57.1%; CI 49.2–64.7) and physical activity also diminished (49.4%; CI 41.6–57.2)

Data were compared between students vs. UE, by the significant difference in the weight variation found between them. Although no significant differences were found on the consumption of vegetables, it is interesting to highlight that it increased by (20.1%) more in the group of students (*p* = 0.062). 

Significant differences were found on the consumption of dairy products, decreasing by (12.9%) in the student’s group (*p* < 0.05); the consumption of pastries and snacks stayed the same, being (30.7%) more in the UE group versus student’s group (*p* < 0.01); consumption of low alcohol drinks (wine and beer), decreased by (31.5%) in the student’s group (*p* < 0.001); physical activity, decreased by (22.5%) in the UE group (*p* < 0.05) (Table 2). Completed data can be found in Appendix A.

Comparing by sex, significant differences were found: women did the grocery 1 time or less per week (16%) more than men (*p* < 0.05); and women increased the consumption of fish/seafood (15.9%) more than men (*p* < 0.05); non-significant differences were found in the remaining questions, although close to significant differences were found in the numbers of meals per day with an increase of (17.9%) in women (*p* = 0.077) (Table 3).

### 3.3. Adherence to the MD (MEDAS-14) during Confinement

Analyzing the adherence to the MD during confinement, (20.8%) of individuals had a low level of adherence to the MD, and Medium/High in 79.2%. 

Specifically, participants with a medium/high adherence to the MD during confinement were men (82.1%; CI 69.6–91.0), born in Spain (84.2%; CI 76.8–89.9) compared to Other Countries (15.7%; CI 10.0–23.1) (*p* < 0.05), students (79.8%; CI 71.8–86.3), living in family home (82.8%; CI 75.1–88.9) compared to non-family home (67.5%; CI 50.8–81.4) (*p* < 0.05), and by the BMI, in under/normal weight (<18.5–24.9 kg/m^2^) (81.3%; CI 73.2–87.7) (Table 4).

Analyzing for students with a medium/high adherence, they were from the Human Nutrition degree course (96%; CI 79.6–99.8) compared to other degrees (75.9%; CI 66.5–83.8) (*p* < 0.05) and from the Faculty of Health Science (80%; CI 69.9–87.8) compared with other faculties (Table 5). 

### 3.4. Emotional Eater Behavior during Confinement

The EEQ survey was used to evaluate the emotional relationship with food intake during lockdown. In our sample, most of the subjects were Non-emotional/Low Emotional eaters (72.6%) compared to Emotional/Very emotional eaters (27.4%).

The profile of participants with an Emotional/Very emotional eating behavior, during confinement, was female (35.7%; CI 26.8–45.3) compared to male (10.7%; 4.0–21.8) (*p* < 0.01), Other Countries (40%; CI 23.8–57.8) compared to Spanish (24.0%; CI 17.0–32.2) (*p* = 0.060), students (29.4%; CI 21.7–38.1), living in non-family home (32.5%; CI 18.5–49.1) and with Pre-obesity/Obesity (37.7%; CI 23.7–53.4) compared to Under/Normal weight (23.5%; CI 16.3–32.0) (*p* = 0.068) (Table 6). According to degree, non-Human Nutrition (30.7%; 22.0–40.5) and from non-Health Sciences (36.3%; CI 22.4–52.2) (Table 7). 

## 4. Discussion

The spread of COVID-19 around the world forced governments to make very restrictive decisions [38]. In the case of Spain, a strict confinement of the entire population was imposed on 14 March 2020, which limited mobility [4]. Being forced to stay indoors for a long time can dramatically change our eating habits and physical activity practices [10,26,39,40]. Lockdown was an unpleasant experience for everyone, especially for those who lived alone [41,42]. Job uncertainty, loneliness and fear of illness [43,44], besides the closure of schools, work activities, social distance and the ban of physical activities, radically disrupted the concept of normality [39,40]. In this work, we have analyzed these modifications in a population sample from the university community of the Universidad Europea del Atlántico located in Santander, Cantabria, Spain. To our knowledge this is the first study that has analyzed differences between staff and students in a Spanish college population and which analyzed how confinement affects the population of students and staff in the same university in terms of eating habits, physical activity and emotional eating. The main findings of this study were the significant differences in weight gain between students and university staff (*p* < 0.05), significant differences in the consumption of dairy products, with a decrease of (12.9%) in the group of students (*p* < 0.05), consumption of pastries and snacks stayed the same, being (30.7%) more in the UE group versus student’s group (*p* < 0.01); consumption of low alcohol drinks (wine and beer), decreased by (31.5%) in the group of students (*p* < 0.05) and physical activity increased by (17.2%) in the group of students (*p* < 0.001). As for adherence to MD (20.8%), our population had a low level of adherence to MD. Finally, (27.4%) of our population were emotional/Very emotional eaters, with significantly higher values in women (*p* < 0.01).

Before lockdown, the prevalence of subjects who were overweight or obese was 26.8% lower compared to the data reported for the Spanish population by Sánchez-Sánchez et al. [11], who found an incidence of 39.8%, and similar to the data reported by Sidor et al. [7] for the Polish population. It should be noted that in this case, the distribution of subjects who were overweight or obese also coincided with those of our sample. The incidence in our sample was also lower compared to the data reported for the Chilean (52.2%) [45] and Dutch (55.6%) [46] populations. This may be due to the average age of our population which was mostly made up of students. In general, weight variations were observed in the participants: 36.3% weight loss and a 35.7% weight gain with significant differences between students and UE respectively. Weight loss was observed more in students than in UE (*p* < 0.05) and no correlation was found between BMI and weight gain/loss. Our results show a percentual lower weight gain in the population than those found in Sinisterra-Loaiza [28], Sidor et al. [7] and Sánchez-Sánchez et al. [11], who reported a weight loss in obese people during confinement as in Di Renzo et al. [27]. However, it is consistent with the weight gain reported in Celorio-Sardá et al. [30]; Reyes-Olavarria et al. [45] and Deschasaux-Tanguy et al. [9]. It is interesting to note that no differences were found between health science students and the rest of the university population.

Regarding the dietary and lifestyle adaptations during lockdown in our populations, the consumption of fruits, vegetables, dairy, pulses, fish/seafood, white meat, red and processed meat, pastries and snacks, rice/pasta/potatoes, nuts, low alcohol drinks, spirits and sugary drinks, number of meals per day and physical activity decreased or stayed the same. The only item that increased was the frequency of cooking during confinement. These results are similar to those found by Rodríguez-Pérez et al. [43] and contrary to those found by Sánchez-Sánchez et al. [11], who reported an increase in the consumption of alcoholic beverages, and Celorio-Sardá et al. [30] who reported an increased consumption in fruit, vegetables, legumes, fish, eggs and yogurt. We can conclude that most of our population did not change their eating habits during confinement, similar to the results found in 1932 Italian adults by Scarmozzino et al. [10] and Brancaccio et al. [26] and by Poelman et al. [46] in 1030 Dutch adults. The decrease in the consumption of pastries and snacks is contrary to what was found in French population by Marty et al. [8].

Because of the significant differences found in weight variation, we analyzed differences between students and UE in dietary and lifestyle adaptations. Significant differences were found in dairy products and low alcohol drinks (where students decreased their consumption more), pastries and snacks (where UE stayed the same) and physical activity (where UE decreased it more). Vegetable consumption also showed an increase, although not significant in the student group. These results are consistent with the results shown by Celorio-Sardá et al. [30] who reported an increased consumption in yogurt and vegetables and decreased consumption in fermented alcoholic beverages, especially in students.

Two key elements of good health are both healthy eating and physical exercise. During the confinement, 36.3% of our population increased their physical activity. Our results are similar, although higher, to those reported by Rodríguez-Pérez et al. [47], Di Renzo et al. [27] and somewhat lower than those found by Celorio-Sardá et al. [30], contrary to those of Romero-Blanco et al. [31] who found that all groups increased their physical activity on average, while in our study significant differences were found between students and UE, who were the ones who decreased their physical activity by the highest percentage (66.7% vs. 44.7%).

During the COVID-19 pandemic, following the MD pattern was recommended due to its ability to stimulate the immune system [19,20]. According to Pérez-Araluce et al. [48], a better adherence to this pattern may be associated with a lower risk of COVID-19. The MD, considered an example of a healthy balanced diet, is characterized by the intake of olive oil, fresh fruits and vegetables [49] which are associated to the reduction of inflammatory stress markers, improvement in lipid profile and insulin sensitivity. These effects are attributable to the phenolic compounds and mono and poly-unsaturated fatty acids they contain [20,50,51]. Our data show that 79.2% of the population had Medium/High adherence to the MD (MEDAS-14) during confinement, which is much higher than the 8% reported by Sánchez-Sánchez et al. [11] for the Spanish population. Comparing by role, UE had the worst results, contrary to what was found by Rodríguez-Pérez et al. [47] and León-Muñoz et al. [51]. Our results could be explained by the fact that among the youngest participants in our study the majority belong to the Faculty of Health Sciences, where they are constantly encouraged to adopt a healthy lifestyle. Among the students, a significant difference was found with a greater adherence to the Mediterranean diet by the students of the Degree in Human Nutrition and Dietetics (96% Moderate/high adherence), which agrees with the data by Celorio-Sardá et al. [30], who found that nutrition students were the ones who most improved their eating habits during confinement. 

The restrictions of confinement, explained above, have been linked to an increase in mental illness, especially depression and anxiety [52]. In this sense, different authors relate an increase in anxiety linked to emotional eating [53,54].

According to the results of the EEQ, most of the participants were categorized as non/low-emotional eaters (72.6%), similar to the data reported by López-Moreno M et al. [55] for Spanish adults (67.2%). Women scored significantly higher than men as emotional/very emotional eaters (35.7% vs. 10.7%) (*p* < 0.01). Similar results were obtained by Di Renzo L et al. [39], who reported that women were more prone to emotional eating than men and no differences were found by age range groups. Social isolation imposed during the COVID 19 pandemic could pose a psychological burden for many individuals, but, for women, who continue to carry the burden of household chores in our current models of society, this led to eating more frequently or in greater quantities as a mechanism for coping with increasing fear and anxiety. It is known that during prolonged stress, our body releases cortisol, which increases the feeling of hunger [56]. 

This study has several limitations. The cross-sectional design does not provide any information on the possible causal nature and has inherent recall error as participants were asked to recall their responses before lockdown and the data obtained depends on the participants’ memory; also, self-reported questionnaires are prone to underreporting and susceptible to biased information. Our sample size is too small to be representative, this causes the results to have more variability which could lead to bias and, therefore, the conclusions cannot be extrapolated. The distribution among genders and age range groups is inadequate. Due to the exceptional nature of the situation and the data to be analyzed, there are no validated questionnaires that capture variation in food consumption during a lockdown situation. The lack of a validated questionnaire can omit relevant information and raise the problem of generalization of results. Future studies should consider whether the effects of these lockdowns has caused long-term consequences on eating and physical behavior. It would also be interesting to validate a questionnaire for emergency isolation situations that could be applied in possible future pandemic situations.

## 5. Conclusions

We can affirm that under the lockdown caused by COVID-19, the UNEAT community decreased or maintained the same dietary and lifestyle habits; students increased the intake of fruits and vegetables and physical activity compared with UE, whereas UE, during lockdown, increased the intake of low alcohol drinks (wine and beer). As mentioned before, fruits and vegetables are the principal source of bioactive compounds that help to maintain a state of health and could be a therapeutic approach to COVID-19. 

In addition, our sample limited the consumption of pastries and snacks, sugary drinks, and alcoholic beverages, had a Medium/High adherence to the MD and practiced physical exercise regularly. This resulted in a reduction or maintenance of weight, especially in the youngest subjects in the community. In an exceptional situation such as a pandemic scenario, it is difficult to establish a bidirectional relationship between foods and mood. In any case, in our sample, women were more likely to be emotional eaters than men, eating for reasons other than hunger, and increasing the intake of more palatable foods.

Future strategies should be implemented to promote a balanced and healthy diet, especially improving the consumption of fruits and vegetables, together with the practice of physical activity, for the entire University population, and for the UE, who increased weight more during lockdown.

## Figures and Tables

**Figure 1 ijerph-19-02210-f001:**
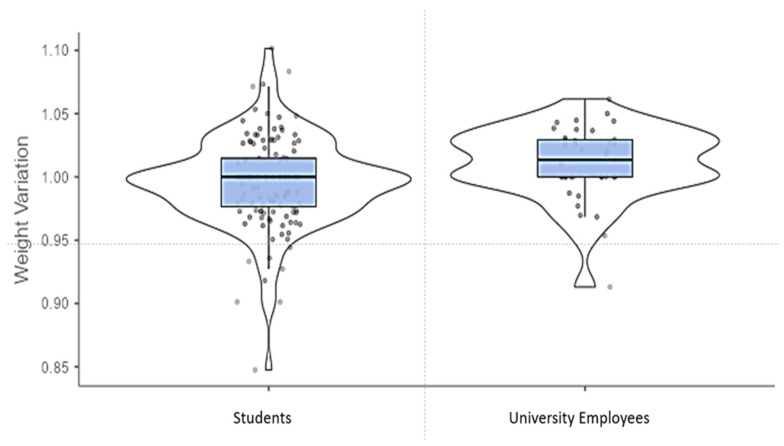
Distribution of weight variation compared by role. Weight change is expressed in weight variation (weight increased during the lockdown in kg/weight before the lockdown in kg); 1 meant that current weight was maintained, <1 implied a weight reduction and >1 weight gain. Differences were evaluated by the Mann–Whitney U test (*p* < 0.05).

**Table 1 ijerph-19-02210-t001:** Sociodemographic characteristics compared by Students vs. UE ^1^.

		Total *N* = 168 (100%; CI)	Students *N* = 129 (76.8%; CI)	UE *N* = 39 (23.2%; CI)	*p*-Value ^2^
Gender					
	Women	112 (66.7%; 58.9–73.7)	92 (71.3%; 62.6–78.9)	20 (51.3%; 34.7–67.5)	<0.05
	Men	56 (33.3%; 26.2–41.0)	37 (28.7%; 21.0–37.3)	19 (48.7%; 32.4–65.2)	
Birthplace					
	Spain	133 (79.2%; 72.2–85.0)	99 (76.7%; 68.4–83.7)	34 (87.2%; 72.5–95.7)	<0.05
	Latin America	28 (16.7%; 11.3–23.1)	26 (20.2%; 13.6–28.1)	2 (5.1%; 0.6–17.3)	
	Europe	4 (2.4%; 0.6–5.9)	1 (0.8%; 0.0–4.2)	3 (7.7%; 1.6–20.8)	
	Others	3 (1.8%; 0.3–5.1)	3 (2.3%; 0.4–6.6)	0 (0%)	
Place of Residence					
	Family home	128 (76.2%; 69.0–82.4)	97 (75.2%; 66.8–82.3)	31 (79.5%; 63.5–90.7)	0.081
	Shared flat	19 (11.3%; 6.9–17.0)	16 (12.4%; 7.2–19.3)	3 (7.7%; 1.6–20.8)	
	Student residence	10 (6%; 2.8–10.6)	10 (7.8%; 3.7–13.7)	0 (0%)	
	Alone	11 (6.5%; 3.3–11.4)	6 (4.7%; 1.7–9.8)	5 (12.8%; 4.2–27.4)	
Age Range					
	<20 years	54 (32.1%; 25.1–39.7)	54 (41.9%; 33.2–50.8)	0 (0%)	<0.001
	21–35 years	80 (47.6%; 39.8–55.4)	73 (56.6%; 47.5–65.2)	7 (17.9%; 7.5–33.5)	
	>36 years	34 (20.2%; 14.4–27.1)	2 (1.6%; 0.1–5.4)	32 (82.1%; 66.4–92.4)	
BMI (kg/m^2^) ^3^					
	Underweight (<18.5 kg/m^2^)	12 (7.1%; 3.7–12.1)	12 (9.3%; 4.9–15.6)	0 (0%)	<0.05
	Normal weight (18.5–24.9 kg/m^2^)	111 (66.1%; 58.3–73.1)	90 (69.8%; 61.0–77.5)	21 (53.8%; 37.1–69.9)	
	Pre-obesity (25–29.9 kg/m^2^)	36 (21.4%; 15.4–28.4)	22 (17.1%; 11.0–24.6)	14 (35.9%; 21.2–52.8)	
	Obesity class I and II (30–34.9 kg/m^2^)	9 (5.4%; 2.4–9.9)	5 (3.9%; 1.2–8.8)	4 (10.3%; 2.8–24.2)	

^1^ UE: University Employees. ^2^ Differences between role were evaluated by the Pearson Chi-square test (*p* < 0.05). ^3^ BMI: Body Max Index, expressed in kg/m^2^, using WHO classification.

**Table 2 ijerph-19-02210-t002:** Dietary and lifestyle adaptations during confinement, comparing Students vs. UE.

	Total *N* = 168 (100%; CI)	Students *N* = 129 (76.8%; CI)	UE ^1^ *N* = 39 (23.2%; CI)	*p*-Value ^2^
During confinement, the consumption of vegetables?				
Has increased	69 (41.1%; 33.5–48.9)	59 (45.7%; 36.9–54.7)	10 (25.6%; 13.0–42.1)	0.062
Has decreased	14 (8.3%; 4.6–13.5)	11 (8.5%; 4.3–14.7)	3 (7.7%; 1.6–20.8)	
Has stayed the same	85 (50.6%; 42.7–58.3)	59 (45.7%; 36.9–54.7)	26 (66.7%; 49.7–80.9)	
During confinement, the consumption of dairy products?				
Has increased	43 (25.6%; 19.1–32.8)	35 (27.1%; 19.6–35.6)	8 (20.5%; 9.2–36.4)	<0.05
Has decreased	21 (12.5%; 7.9–18.4)	20 (15.5%; 9.7–22.9)	1 (2.6%; 0.0–13.4)	
Has stayed the same	104 (61.9%; 54.1–69.2)	74 (57.4%; 48.3–66.0)	30 (76.9%; 60.6–88.8)	
During confinement, the consumption of pastries and snacks?				
Has increased	36 (21.4%; 15.4–28.4)	31 (24%; 16.9–32.3)	5 (12.8%; 4.2–27.4)	<0.01
Has decreased	51 (30.4%; 23.5–37.9)	45 (34.9%; 26.7–43.7)	6 (15.4%; 5.8–30.5)	
Has stayed the same	81 (48.2%; 40.4–56.0)	53 (41.1%; 32.5–50.0)	28 (71.8%; 55.1–84.9)	
During confinement, the consumption of low alcohol drinks (wine and beer)?				
Has increased	18 (10.7%; 6.4–16.4)	9 (7%; 3.2–12.8)	9 (23%; 11.1–39.3)	<0.001
Has decreased	101 (60.1%; 52.2–67.5)	87 (67.4%; 58.6–75.4)	14 (35.9%; 21.2–52.8)	
Has stayed the same	49 (29.2%; 22.4–36.6)	33 (25.6%; 18.3–34.0)	16 (41%; 25.5–57.9)	
During confinement, your physical activity?				
Has increased	61 (36.3%; 29.0–44.0)	52 (40.3%; 31.7–49.3)	9 (23.1%; 11.1–39.3)	<0.05
Has decreased	83 (49.4%; 41.6–57.2)	57 (44.2%; 35.4–53.1)	26 (66.7%; 49.7–80.9)	
Has stayed the same	24 (14.3%; 9.3–20.5)	20 (15.5%; 9.7–22.9)	4 (10.3%; 2.8–24.2)	

^1^ UE: University Employees. ^2^ Differences between role were evaluated by the Pearson Chi-square test (*p* < 0.05).

**Table 3 ijerph-19-02210-t003:** Dietary and lifestyle adaptations during confinement, compared by gender.

	Total *N* = 168 (100%; CI)	Women *N* = 112 (66.6%; CI)	Men *N* = 56 (33.3%; CI)	*p*-Value ^1^
How many times do you do the grocery shopping per week?				
1 time or less per week	129 (76.8%; 69.6–82.9)	92 (82.1%; 73.7–88.7)	37 (66.1%; 52.1–78.1)	<0.05
2 times or more per week	39 (23.2%; 17.0–30.3)	20 (17.9%; 11.2–26.2)	19 (33.9%; 21.8–47.8)	
During confinement, the consumption of fish/seafood?				
Has increased	39 (23.2%; 17.0–30.3)	32 (28.6%; 20.4–37.8)	7 (12.7%; 5.1–24.0)	<0.05
Has decreased	32 (19%; 13.4–25.8)	18 (16.1%; 9.8–24.2)	14 (25%; 14.3–38.3)	
Has stayed the same	97 (57.7%; 49.8–65.3)	62 (55.4%; 45.6–64.7)	35 (62.5%; 48.5–75.0)	
Have you increased the number of meals these days?				
Has increased	62 (36.9%; 29.6–44.6)	48 (42.9%; 33.5–52.5)	14 (25%; 14.3–38.3)	0.077
Has decreased	84 (50%; 42.2–57.7)	51 (45.5%; 36.0–55.2)	33 (58.9%; 44.9–71.9)	
Has stayed the same	22 (13.1%; 8.3–19.1)	13 (11.6%; 6.3–19.0)	9 (16.1%; 7.6–28.3)	

^1^ Differences between genders were evaluated by the Pearson Chi-square test (*p* < 0.05).

**Table 4 ijerph-19-02210-t004:** Adherence to MD (MEDAS14) during confinement, according to the sociodemographic characteristics of the population.

		Total *N* = 168 (100%; CI)	Low *N* = 35 (20.8%; CI)	Medium/High *N* = 133 (79.2%; CI)	*p*-Value ^1^
Gender					
	Women	112 (66.7%; 58.9–73.7)	25 (22.3%; 14.9–31.1)	87 (77.6%; 68.8–85.0)	0.502
	Men	56 (33.3%; 26.2–41.0%)	10 (17.8%; 8.9–30.3)	46 (82.1%; 69.6–91.0)	
Birthplace					
	Spain	133 (79.2%; 72.2–85.0)	21 (15.7%; 10.0–23.1)	112 (84.2%; 76.8–89.9)	<0.05
	Other Countries	35 (20.8%; 14.9–27.7)	14 (40%; 23.8–57.8)	21 (60%; 42.1–76.1)	
Students and university staff					
	Students	129 (76.8%; 69.6–82.9)	26 (20.1%; 13.6–28.1)	103 (79.8%; 71.8–86.3)	0.694
	UE ^2^	39 (23.2%; 17.0–30.3)	9 (23.0%; 11.1–39.3)	30 (76.9%; 60.6–88.8)	
Place of Residence					
	Family home	128 (76.2%; 69.0–82.4)	22 (17.1%; 11.0–24.8)	106 (82.8%; 75.1–88.9)	<0.05
	Non-family home (shared flat and student residence)	40 (23.8%; 17.5–30.9)	13 (32.5%; 18.5–49.1)	27 (67.5%; 50.8–81.4)	
BMI (kg/m^2^) ^3^					
	Under/Normalweight (<18.5–24.9 g/m^2^)	123 (73.2%; 65.8–79.7)	23 (18.6%; 12.2–26.7)	100 (81.3%; 73.2–87.7)	0.260
	Pre-obesity/Obesity (25–39.9 kg/m^2^)	45 (26.8%;20.2–34.1)	12 (26.6%; 14.6–41.9)	33 (73.3%; 58.0–85.3)	

^1^ Differences of MEDAS-14, were evaluated by the Pearson Chi-square test (*p* < 0.05). ^2^ UE: University Employees. ^3^ BMI: Body Max Index, expressed in kg/m^2^, using WHO classification.

**Table 5 ijerph-19-02210-t005:** Adherence to MD (MEDAS14) during confinement, according to degree and faculty.

		Total *N* = 129 (100%; CI)	Low *N* = 26 (20.2%; CI)	Medium/High *N* = 103 (79.8%; CI)	*p*-Value ^1^
Degree					
	Human Nutrition	25 (19.4%; 12.9–27.2)	1 (4.0%; 0.1–20.3)	24 (96%; 79.6–99.8)	<0.05
	Other degrees	104 (80.6%; 72.7–87.0)	25 (24.0%; 16.2–33.4)	79 (75.9%; 66.5–83.8)	
Faculty					
	Health Sciences	85 (65.9%; 57.0–74.0)	17 (20%; 12.1–30.0)	68 (80%; 69.9–87.8)	0.951
	Other faculties	44 (34.1%; 25.9–42.9)	9 (20.4%; 9.8–35.3)	35 (79.5%; 64.6–90.1)	

^1^ Differences of MEDAS-14, were evaluated by the Pearson Chi-square test (*p* < 0.05).

**Table 6 ijerph-19-02210-t006:** EEQ ^1^ during confinement, according to the sociodemographic characteristics of the population.

		Total *N* = 168 (100%; CI)	Non-Emotional/ Low Emotional Eater *N* = 122 (72.6%; CI)	Emotional/ Very Emotional Eater *N* = 46 (27.4%; CI)	*p*-Value ^2^
Gender					
	Women	112 (66.7%; 58.9–73.7)	72 (64.2%; 54.6–73.1)	40 (35.7%; 26.8–45.3)	<0.01
	Men	56 (33.3%; 26.2–41.0%)	50 (89.2%; 78.1–95.9)	6 (10.7%; 4.0–21.8)	
Birthplace					
	Spain	133 (79.2%; 72.2–85.0)	101 (75.9%; 67.7–82.9)	32 (24.0%; 17.0–32.2)	0.060
	Other Countries	35 (20.8%; 14.9–27.7)	21 (60%; 42.1–76.1)	14 (40%; 23.8–57.8)	
Students and university staff					
	Students	129 (76.8%; 69.6–82.9)	91 (70.5%; 61.8–78.2)	38 (29.4%; 21.7–38.1)	0.272
	UE ^3^	39 (23.2%; 17.0–30.3)	31 (79.4%; 63.5–90.7)	8 (20.5%; 9.2–36.4)	
Place of Residence					
	Family home	128 (76.2%; 69.0–82.4)	95 (74.2%; 65.7–81.5)	33 (25.7%; 18.4–34.2)	0.406
	Non-family home (shared flat and student residence)	40 (23.8%; 17.5–30.9)	27 (67.5%; 50.8–81.4)	13 (32.5%; 18.5–49.1)	
BMI (kg/m^2^) ^4^					
	Under/Normalweight (<18.5–24.9 g/m^2^)	123 (73.2%; 65.8–79.7)	94 (76.4%; 67.9–83.6)	29 (23.5%; 16.3–32.0)	0.068
	Pre-obesity/Obesity (25–39.9 kg/m^2^)	45 (26.8%;20.2–34.1)	28 (62.2%; 46.5–76.2)	17 (37.7%; 23.7–53.4)	

^1^ EEQ: Emotional Eater Questionnaire. ^2^ Differences of EEQ, were evaluated by the Pearson Chi-square test (*p* < 0.05). ^3^ UE: University Employees. ^4^ BMI: Body Max Index, expressed in kg/m^2^, using WHO classification.

**Table 7 ijerph-19-02210-t007:** EEQ ^1^ during confinement, according to degree and faculty.

		Total *N* = 129 (100%; CI)	Non-Emotional/ Low Emotional Eater *N* = 91 (70.5%; CI)	Emotional/ Very Emotional Eater *N* = 38 (29.5%; CI)	*p*-Value ^2^
Degree					
	Human Nutrition	25 (19.4%; 12.9–27.2)	19 (76%; 54.8–906)	6 (24%; 9.3–45.1)	0.505
	Others	104 (80.6%; 72.7–87.0)	72 (69.2%; 59.4–77.9)	32 (30.7%; 22.0–40.5)	
Faculty					
	Health Sciences	84 (65.6%; 56.2–73.2)	63 (75%; 64.3–83.8)	22 (26.1%; 17.1–36.9)	0.216
	Others	44 (34.4%; 25.9–42.9)	28 (63.6%; 47.7–77.5)	16 (36.3%; 22.4–52.2)	

^1^ EEQ: Emotional Eater Questionnaire. ^2^ Differences of EEQ, were evaluated by the Pearson Chi-square test (*p* < 0.05).

## Data Availability

Not applicable.

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
