# Peer review of "Changes in the Lifestyle of the Spanish University Population during Confinement for COVID-19"

_ijerph, 2022, doi:10.3390/ijerph19042210_

Round 1

Author Response

Dear reviewer,

First of all, we would like to thank the reviewer for his advice and recommendations which have allowed us to improve the article. We will now proceed to respond point by point to his recommendations. Corrections have been made in red. 

Material and Methods

  • The national legislation on data protection is not updated, there is a more current law than the one mentioned.
  • The laws must be accompanied by their corresponding bibliographic reference.

Thank you for your indication, we have proceeded to update the indicated law and to include its bibliographical reference and that of the rest of the laws in the bibliography.

  • The authors mention having used the non-parametric Mann-Whitney U test, so it follows that the population does not follow a normal distribution. In these cases, the results are often expressed by the median accompanied by their interquartile range (IQR).

The recommendation is appreciated and the IQRs have been included.

  • It would be appropriate that the categorical variables represented in terms of frequency distribution are accompanied by their corresponding confidence intervals.

The recommendation is appreciated and the confidence intervals have been included in categorical variables represented in terms of frequency distribution.

Results

  • The Results section is too large. It would be appropriate to summarizer the main results.

Following your recommendation, some of the information previously provided in the results has been included as supplementary material.

Discussion.

  • The first paragraph of the discussion section usually includes a summary with the main findings obtained in the research.

Thank you for your suggestion, we have included a summary with the main findings in the research in the first paragraph of the discussion.

  • The sample size is small. This aspect is mentioned by the authors in the limitations, but this reviewer considers that further discussion of this important limitation of the study is necessary.

Following your suggestion, a more in-depth discussion of the simple size limitations have been included.

  • The structure of the abstract does not follow the recommendations of the journal. The abstract should be a single paragraph and should follow the style of structured abstracts, but without headings.

Thank you for your comment, the summary has been improved.

  • Figure 1 does not appear in the manuscript

We regret the error and figure 1 has been included.

  • It would be advisable to review the bibliography, since, at least, reference 4 and 5 are wrong.

Thank you for your recommendation, the bibliography has been checked to ensure that all references are correctly cited.

Reviewer 2 Report

The authors conducted a cross-sectional design, online survey to evaluate the influence of a COVID-19 lockdown pandemic on lifestyle behaviours in a university convenance sample. The majority of the manuscript is adequately detailed, however there are numerous corrections that are required, as shown below.

Abstract lines 29-36: The results section should be reworded to improve clarity. Description provided is not specific e.g. fruit consumption “decreased or stayed the same” – which is correct? Since the study aim is to evaluate the influence of lockdown, all findings presented should clearly indicate influence of change from lockdown. Also, check statistical values are correct (e.g. should line 38 be p<0.001 instead of p<0.01?)

Lines 71-78, this is very long sentence and should be reworded to improve clarity and grammar.

Line 95, this statement should be rephrased towards a lack of studies focusing on university populations in Spain specifically, as there are numerous publications globally pertaining to lifestyle behavior changes within university populations with larger sample sizes than the present study.

Line 99, sentence should be reworded as other studies have indeed investigated emotional eating during lockdown, one example here https://doi.org/10.1016/j.appet.2021.105122

Line 127, it should be clearly described in this section how questions were formulated to ascertain responses pertaining to before and during lockdown. What was the length of recall that was asked of participants to recall in the questionnaire?

On line 129, describe in more detail what questions were used to determine body weight, how were participants instructed to record weight? Further, was height determined as this is essential for BMI calculation and was not stated.

An important criterion for selecting questionnaires for lifestyle behaviors is the evidence of validity and reliability. Regarding lines 133-139, please clarify if questions used were derived from previously validated questionnaires and provide detail/citation support on these. If not, strong rationale needs to be provided for why these were not used, as various options are available and widely utilized in the literature base.

Line 138, how was moderate and vigorous physical activity defined to participants in the questionnaire? Was a validated questionnaire such as the IPAQ used?

Line 151, how were any survey responses with missing data incorporated? Were they included or excluded from analysis?

Line 182 (Table 1), since a sub analysis was conducted comparing University students and employees, sociodemographic characteristics should be provided for each population group – similar to how you have split the groups in in Table 3.

Line 188-189, this description would be more appropriately placed in the methods section around line 139.

Line 214, was shopping frequency determined also for before lockdown or only during lockdown?

Line 230 (Table 3), check decimal placing of data on this table, ensure data are expressed in a consistent manner. Same advice applies to your other tables.

Line 272, this is a broad statement that should have more rationale added.

Line 277-279, as previously noted this needs to be amended as previous studies have studied this topic.

Line 319, this statement lacks validity the way it is phrased – I recommend rewording it. Physical activity and healthy diet each individually provide a health effect, and collectively the healthy effect is magnified.

Line 364, further limitations should be emphasized. On line 360, the single cross-sectional design also has inherent recall error as participants were asked to recall their responses for before lockdown. Lines 362-364, in regards to the validity of questionnaires, provide rationale as to why validated questionnaires were not used and indicate recommendations for future research regarding this.

Author Response

Dear reviewer,

First of all, we would like to thank the reviewer for his advice and recommendations which have allowed us to improve the article. We will now proceed to respond point by point to his recommendations. Corrections have been made in red. 

Best regards, 

Abstract lines 29-36: The results section should be reworded to improve clarity. description provided is not specific e.g. fruit consumption “decreased or stayed the same” – which is correct? Since the study aim is to evaluate the influence of lockdown, all findings presented should clearly indicate influence of change from lockdown. Also, check statistical values are correct (e.g. should line 38 be p<0.001 instead of p<0.01?)

Thank you for your comment, the summary has been improved.

Lines 71-78, this is very long sentence and should be reworded to improve clarity and grammar.

Thank you for your comment, the sentence has been improved.

Line 95, this statement should be rephrased towards a lack of studies focusing on university populations in Spain specifically, as there are numerous publications globally pertaining to lifestyle behaviour changes within university populations with larger sample sizes than the present study.

The suggestion is appreciated and the sentence has been revised.

Line 99, sentence should be reworded as other studies have indeed investigated emotional eating during lockdown, one example here https://doi.org/10.1016/j.appet.2021.105122

The suggestion is appreciated and the sentence has been revised.

Line 127, it should be clearly described in this section how questions were formulated to ascertain responses pertaining to before and during lockdown. What was the length of recall that was asked of participants to recall in the questionnaire?

The comment is appreciated and the requested information has been included.

On line 129, describe in more detail what questions were used to determine body weight, how were participants instructed to record weight? Further, was height determined as this is essential for BMI calculation and was not stated.

The comment is appreciated and the requested information has been included.

An important criterion for selecting questionnaires for lifestyle behaviors is the evidence of validity and reliability. Regarding lines 133-139, please clarify if questions used were derived from previously validated questionnaires and provide detail/citation support on these. If not, strong rationale needs to be provided for why these were not used, as various options are available and widely utilized in the literature base.

We are grateful for the suggestion, but it should be borne in mind that, given that what we want to measure is the change in eating habits before and during confinement, there is no validated questionnaire to collect this information, so, as other authors have done (https://doi.org/10.3390/ijerph17155431, https://doi.org/10.3390/nu13051494), a specific questionnaire had to be drawn up.

To complement the information collected and to make it more robust, two validated questionnaires, the Medas 14 (doi:10.3945/jn.110.1355) and the EEQ (doi:10.3305/nh.2012.27.2.5659)

Line 138, how was moderate and vigorous physical activity defined to participants in the questionnaire? Was a validated questionnaire such as the IPAQ used?

Thank you for your comment. It was an error in the description, because the participants were only asked whether their activity had decreased, remained the same or increased during confinement.

Line 151, how were any survey responses with missing data incorporated? Were they included or excluded from analysis?

The comment is appreciated and the requested information has been included.

Line 182 (Table 1), since a sub analysis was conducted comparing University students and employees, sociodemographic characteristics should be provided for each population group – similar to how you have split the groups in in Table 3.

The comment is appreciated and the requested information has been included.

Line 188-189, this description would be more appropriately placed in the methods section around line 139.

The suggestion is gratefully acknowledged and the information has been moved.

Line 214, was shopping frequency determined also for before lockdown or only during lockdown?

The comment is appreciated and the requested information has been included.

Line 230 (Table 3), check decimal placing of data on this table, ensure data are expressed in a consistent manner. Same advice applies to your other tables.

Thank you for the recommendation, all tables have been revised.

Line 272, this is a broad statement that should have more rationale added.

Thank you for your suggestion, the paragraph has been amended and expanded.

Line 277-279, as previously noted this needs to be amended as previous studies have studied this topic.

The suggestion is appreciated and the sentence has been revised.

Line 319, this statement lacks validity the way it is phrased – I recommend rewording it. Physical activity and healthy diet each individually provide a health effect, and collectively the healthy effect is magnified.

The suggestion is appreciated and the sentence has been revised.

Line 364, further limitations should be emphasized. On line 360, the single cross-sectional design also has inherent recall error as participants were asked to recall their responses for before lockdown. Lines 362-364, in regards to the validity of questionnaires, provide rationale as to why validated questionnaires were not used and indicate recommendations for future research regarding this.

Thanks for the suggestion, the limitations of the study have been expanded and recommendations for future research have been included.

Reviewer 3 Report

This study investigated the impacts of lockdown on the lifestyle of university population during COVID-19. It is an important topic; however, the current version needs a major revision.

  1. In table 3, "has decreased" and "stay the same" need to be separated. It is critical to know how much people "has decreased" and how much people "stay the same".
  2. The detailed data needs to be provided as the supporting information.  

Author Response

Dear reviewer,

First of all, we would like to thank the reviewer for his advice and recommendations which have allowed us to improve the article. We will now proceed to respond point by point to his recommendations. Corrections have been made in red. 

Best regards, 

  1. In table 3, "has decreased" and "stay the same" need to be separated. It is critical to know how much people "has decreased" and how much people "stay the same".

Thank you for your suggestion, the indicated changes have been made.

  1. The detailed data needs to be provided as the supporting information. 

Thank you for your suggestion. On your advice, some data have been included as supporting information.

Round 2

Reviewer 1 Report

The quality of the manuscript has improved and is suitable for publication.

However, a final minor revision is suggested.

As suggested by this reviewer, the authors included findings expressed by the median accompanied by the corresponding interquartile range.

Therefore, it would be advisable to indicate that this parameter is to be used in the statistical analysis section.

For example:

Lines 164-165.

The quantitative variables were expressed as mean and standard deviation (SD) or as median accompanied by their interquartile range (IQR) as appropriate.

Author Response

Dear Reviewer,

First of all, we would like to thank you for your help in improving our article. Below we respond to your feedback.

Therefore, it would be advisable to indicate that this parameter is to be used in the statistical analysis section.

For example:

Lines 164-165.

The quantitative variables were expressed as mean and standard deviation (SD) or as median accompanied by their interquartile range (IQR) as appropriate.

Thank you for your advice, we have included the recommended text in the material and methods section.

Best regards, 

Reviewer 2 Report

The authors have made appropriate amendments in relation to previous feedback which has strengthened the manuscript. Further English language proof reading is recommended throughout the manuscript to correct present grammatical errors and to improve flow. Please also ensure abbreviations used in tables and figures are defined below each table/figure.

Author Response

Dear Reviewer,

First of all, we would like to thank you for your help in improving our article. Below we respond to your feedback.

The authors have made appropriate amendments in relation to previous feedback which has strengthened the manuscript. Further English language proof reading is recommended throughout the manuscript to correct present grammatical errors and to improve flow.

A review has been carried out to correct present grammatical errors and to improve flow

Please also ensure abbreviations used in tables and figures are defined below each table/figure.

All tables and figures have been checked to ensure that all abbreviations are defined below them.

Best regards, 

Reviewer 3 Report

The authors have addressed my comments. There is just one minor revision needs to be made. In Table 2, for the responses of "the consumption of diary products", there are two "has stay the same". 

Author Response

Dear Reviewer,

First of all, we would like to thank you for your help in improving our article. Below we respond to your feedback.

The authors have addressed my comments. There is just one minor revision needs to be made. In Table 2, for the responses of "the consumption of dairy products", there are two "has stay the same".

Thank you for your revision, table 2 have been corrected.

Best regards,